# Sleep Disturbances in Children with Attentional Deficit Hyperactivity Disorder and Specific Learning Disorders

**DOI:** 10.3390/ijerph19116411

**Published:** 2022-05-25

**Authors:** Maria Silvia Saccani, Luciana Ursumando, Silvia Di Vara, Giulia Lazzaro, Cristiana Varuzza, Stefano Vicari, Deny Menghini

**Affiliations:** 1Child and Adolescent Neuropsychiatry Unit, Department of Neuroscience, Bambino Gesù Children’s Hospital, IRCCS, 00146 Rome, Italy; mariasilvia.saccani@phd.unipd.it (M.S.S.); luciana.ursumando@opbg.net (L.U.); silvia.divara@opbg.net (S.D.V.); giulia.lazzaro@opbg.net (G.L.); cristiana.varuzza@opbg.net (C.V.); stefano.vicari@opbg.net (S.V.); 2Department of General Psychology, Padova Neuroscience Center, University of Padua, 35122 Padua, Italy; 3Department of Life Sciences and Public Health, Catholic University, 00168 Rome, Italy

**Keywords:** neurodevelopmental disorders, ADHD, dyslexia, insomnia

## Abstract

Sleep disturbances may be a significant source of distress for children with neurodevelopmental disorders, and consequently also for their families. Crucially, sleep disturbances might be influenced by comorbidity. Attention deficit hyperactivity disorder (ADHD) and specific learning disorder (SLD) often co-occur, and consequently, investigating sleep disturbances in children with comorbidity of ADHD and SLD is essential. Our study aimed at detecting sleep difficulties in a group of 74 children with ADHD, 78 children with SLD, and 76 children with ADHD and SLD by using the Sleep Disturbances Scale for Children. The results showed that sleep difficulties emerge more clearly in children with comorbid ADHD and SLD compared to children with only ADHD or SLD. These sleep difficulties were not due to differences in ages and behavioral/emotional problems. In conclusion, evaluating sleep disturbances is important when assessing and managing children with ADHD, SLD, and particularly with the two comorbid conditions, to better understand their difficulties and develop tailored interventions.

## 1. Introduction

Sleep disturbances are common in children with neurodevelopmental disorders and might become a significant source of distress, compromising the quality of life of both children and families [1]. Insufficient sleep is associated with emotional problems, such as depressed mood and anxiety traits, behavioral problems, such as impulsivity, hyperactivity, and aggressiveness, and cognitive problems, such as memory difficulties, learning difficulties, reduced attention, and reduced executive functioning [2,3,4].

Sleep disturbances have been unevenly studied in neurodevelopmental disorders. In children with attention deficit hyperactivity disorder (ADHD), 55% to 74% of sleep difficulties have been documented [5,6], such as decreased sleep efficiency, daytime sleepiness, delayed sleep onset, difficulty falling asleep, bedtime resistance problems, sleep-related breathing problems, motor restlessness during sleep, nocturnal enuresis, nocturnal awakenings, and difficulty waking up in the morning [7,8,9,10]. In addition, it has been noted that ADHD-like behavioral and cognitive symptoms may actually be consequences of sleep difficulties [11,12,13].

However, findings documenting atypical sleep and reduced sleep quality in children with ADHD are controversial [3,14,15]. Among the confounding factors, the most important are [7,8]: the greater intra-individual variability of the sleep/wake cycle in children with ADHD compared with controls, even when controlling for gender, medication, number of days with data, and social jetlag [16]; age groups, as younger children with ADHD tend to have shorter sleep and older children longer sleep than controls [17]; and ADHD presentation, as children with ADHD with combined presentation show greater sleep problems than controls in initiating and maintaining sleep, breathing, sleep–wake transition, excessive daytime sleepiness, and sleep hyperhidrosis [18]. Discrepancies in results may also be due to the measures used. Most studies have used subjective measures, such as questionnaires, and only a few studies have employed objective measures, such as polysomnography or actigraphy [7].

When sleep measures were collected via actigraphy [19], children with ADHD presented similar sleep duration to controls and only moderately altered sleep patterns compared to controls, with increased activity during sleep. Children with ADHD who underwent a comprehensive sleep assessment [12,13] using multiple methodologies (e.g., blood tests, sleep questionnaires, laboratory video-polysomnographic recordings, multiple sleep latency tests, and one-week actigraphy) showed a much higher percentage of sleep problems than previous studies (of 30 children, 28 had poor sleep quality) and multiple sleep phenotypes, such as narcolepsy, delayed sleep onset insomnia, obstructive sleep apnea, periodic limb movements, and epileptiform discharges in sleep. In general, when studies controlled for various confounding factors, children with ADHD did not differ from controls in sleep patterns and had only alterations in sleep quality, such as greater daytime sleepiness, more motor restlessness during sleep, and more breath-related sleep disturbances than controls [8].

Unlike ADHD, sleep difficulties in children with specific learning disorders (SLD) have been overlooked. In the pioneering study by Mercier et al. [20], the sleep patterns of children with reading difficulties were measured for four consecutive nights using polysomnography. The results showed that children with reading difficulties, compared to controls, had an increase in stage 4 sleep, a decrease in sleep with rapid eye movements (REM sleep), a delayed onset of REM sleep, and a prolonged initial cycle of non-REM sleep [20].

Bruni et al. [21] measured sleep patterns by recording brain activity with EEG and analyzed spectral power in children with reading difficulties. The results showed that children with reading difficulties had increased EEG power of the frequency bands between 0.5–3 and 11–12 Hz (slow sigma) in stage 2 and between 0.5–1 Hz in stage 3. In addition, an increase in spindle density was found during stage 2. Importantly, these indices were correlated with the degree of reading impairment. In more recent studies, the association between sleep patterns and learning problems in children with reading difficulties has also been demonstrated [22,23].

Sleep disorders may be exacerbated in the case of comorbidity [9,24,25,26,27]. For example, ADHD and Tic disorder have been shown to be characterized by specific sleep difficulties that add up in comorbidity [25]. In addition, several studies have reported that symptoms of depression and anxiety contribute to sleep difficulties in ADHD, regardless of the severity of ADHD symptoms [9,26,27]. Finally, ADHD in combination with oppositional defiant disorder has been found to be associated with increased resistance to going to bed and waking up in the morning [24].

Sleep disorders have never been investigated for comorbidity between ADHD and SLD, despite the two disorders having a very high co-occurrence [28,29] with comorbidity rates ranging from 8% to 76% [30] and resulting in attention problems more severe than ADHD alone and learning problems more severe than SLD alone [31].

The present study aimed to investigate sleep disturbances in ADHD and SLD and to clarify how sleep disturbances are affected by ADHD and SLD comorbidity. To this end, we compared a group of children with ADHD and SLD comorbidity (ADHD+SLD), a group with ADHD, and a group with SLD on the scores of the Sleep Disturbances Scale for Children (SDSC, [32]).

## 2. Materials and Methods

### 2.1. Participants

Children and adolescents with ADHD+SLD, ADHD, and SLD were retrospectively selected on the basis of age and IQ from a large database. This database consisted of several hundred outpatients with behavioral problems and/or learning difficulties referred to the Child and Adolescent Neuropsychiatry Unit of the Bambino Gesù Children’s Hospital in Rome for in-depth diagnostic investigation. All participants were evaluated between 2017 and 2020 by expert neuropsychiatrists and developmental neuropsychologists.

The criteria for inclusion in the study consisted of having a diagnosis of ADHD and/or SLD (reading disorder and/or math disorder and/or writing disorder) according to DSM-5 [33] criteria and national guidelines or recommendations [34]; while the exclusion criteria were as follows: (1) having an intellectual disability or autism spectrum disorder according to DSM-5 [33] criteria; (2) having a personal history of neurological or neurosensory disorders; and (3) having been on drug treatment for at least 6 months.

The diagnosis of ADHD was made according to DSM-5 criteria [33] based on developmental history and an extensive clinical examination. To obtain separate information from children and parents about ADHD symptoms, the semi-structured interview, Schedule for Affective Disorders and Schizophrenia for School-Age Children-Present and Lifetime Version [35], was also administered by an experienced clinician.

The diagnosis of SLD, reading disorder, and/or math disorder and/or writing disorder was diagnosed when the performance (level of accuracy or speed) was at least 2 SDs below the mean for school-age on norm-referenced measures [36,37,38,39,40,41,42,43,44].

The final sample consisted of: 74 participants with ADHD (12 females and 62 males; mean age ± SD: 9.5 ± 2.3 years; age range: 6.4–15 years), 78 participants with SLD (35 females and 43 males; mean age ± SD: 9.89 ± 1.86 years; age range: 7.4–14.6 years), and 76 participants with ADHD+SLD (17 females and 59 males; mean age ± SD: 10 ± 2 years; age range: 7–14.8 years). 

The IQ scores were obtained using the Wechsler Intelligence Scale for Children-IV [45] or the colored/standard progressive matrices [46] as follows: ADHD group, mean IQ ± SD: 112.57 ± 12.84; SLD group, mean IQ ± SD: 108.81 ± 10.32; ADHD+SLD group, mean IQ ± SD: 109.05 ± 12.8.

In accordance with matching criteria, children with ADHD+SLD, ADHD, and SLD did not differ for chronological age (F_2,225_ = 1.01, *p* = 0.36, ηp^2^ = 0.01) and for IQ (F_2,225_ = 2.31, *p* = 0.11, η_p_^2^ = 0.02).

Participants’ anonymity and data confidentiality were ensured. All procedures performed in the study involving human participants were in accordance with the 1964 Declaration of Helsinki and its subsequent amendments or comparable ethical standards. Ethical review and approval were waived for this study because of the retrospective design and the fact that the data used were anonymous. 

### 2.2. Measures of Sleep Disturbances 

The Sleep Disturbances Scale for Children (SDSC, [31]) is an easy, short, and well-validated questionnaire completed by parents and used to assess sleep difficulties over the past 6 months in children and adolescents. The SDSC consists of 26 items with values ranging from 1 to 5 (higher numerical values reflect greater symptom severity). Items are grouped into six disorder subscales: disorders in initiating and maintaining sleep (DIMS), sleep breathing disorders (SBD), disorders of arousal (DA), sleep–wake transition disorder (SWTD), disorders of excessive somnolence (DES), and nocturnal hyperhidrosis (NH).

The SDSC provides a standardized measure of sleep disturbance in childhood and adolescence through a sleep index score. The development of the SDSC was based on a database of a large population in order to define normal values and identify children with disturbed sleep [32]. Therefore, raw scores are converted to T-scores based on normative data. T-scores are clinically significant when they are ≥70, whereas 60 to 70 are considered borderline scores. To have a measure of sleep disturbance, we analyzed the results of all six subscales of the disorder.

### 2.3. Measure of Behavioral and Emotional Symptoms

The Child Behavior Checklist for Ages 6–18 [47] is a widely used questionnaire completed by parents and used to identify behavioral and emotional problems over the past 2 months in children and adolescents. Raw scores were converted to T-scores based on normative data. We analyzed the results of two broadband scales: internalizing problems and externalizing problems.

### 2.4. Statistical Analyses

A Chi-square test for association with the odd ratios (ORs) and 95% confidence intervals (95% CI) was conducted to compare the number of participants with clinical scores of each group (ADHD+SLD, ADHD, SLD) in each SDSC subscale (DIMS, SBD, DA, SWTD, DES, NH). To take into account multiple comparisons (3 Group × 6 SDSC Subscale), the Benjamini–Hochberg procedure was adopted (False Discovery Rate was set at 0.1).

A repeated-measure analysis of variance (RMANOVA) with the Group (ADHD, SLD, ADHD+SLD) as between-subject factors and SDSC Subscales (DIMS, SBD, DA, SWTD, DES, NH) as within-subject factors was run to test whether scores of SDSC subscales differed among groups.

Another RMANOVA with the Group (ADHD, SLD, ADHD+SLD) as the between-subject factor and CBCL broadband scales (Internalizing Problems and Externalizing Problems) as the within-subject factor was run to verify whether the scores of CBCL broadband scales differed between groups.

For both RMANOVAs, post hoc analyses were conducted by means of Tukey’s HSD tests. Partial eta squares (ηp^2^) were reported as measures of the effect size.

Additionally, Pearson’s correlation was used to measure the relation between scores of SDSC subscales (DIMS, SBD, DA, SWTD, DES, NH) and scores of CBCL broadband scales (internalizing problems and externalizing problems). To take into account multiple comparisons (2 CBCL Broadband Scale × 6 SDSC Subscales), Bonferroni correction was applied (*p* = 0.05/12 = 0.004). 

Finally, in order to take into account the possible influence of emotional and behavioral symptoms in comparing the three groups on sleep disturbances, a repeated-measure analysis of variance controlling for CBCL broadband scales scores (RMANCOVA) was run, with the Group (ADHD, SLD, ADHD+SLD) as between-subject factors, the SDSC subscale (DIMS, SBD, DA, SWTD, DES, NH) as within-subject factors, and the CBCL broadband scale as the covariate.

## 3. Results

As shown in Table 1, the number of children with ADHD+SLD who obtained clinical scores in DIMS, DA, SWTD, and DES subscales was higher than that of children with SLD. Furthermore, children with ADHD+SLD compared to children with ADHD showed higher clinical scores in SWTD and NH subscales. Conversely, the comparisons between children with ADHD and children with SLD in all SDSC subscales were not significant.

By comparing the three groups on SDSC subscales scores, a significant main effect of Group (F_2,225_ = 11.43, *p* = 0.001, ηp^2^ = 0.09) was found. Post hoc analyses (Tukey’s HSD tests) showed that children with ADHD+SLD had significantly higher scores than children with ADHD (*p* = 0.011) and children with SLD (*p* = 0.001). Coversely, children with ADHD and children with SLD did not differ (*p* = 0.157) (Figure 1).

A significant main effect of the SDSC subscale was also found (F_5,1125_ = 22.66, *p* = 0.001, ηp^2^ = 0.09). As shown in Table 2, scores of the DIMS subscale were significantly higher than those obtained in the other subscales. Instead, the Group × SDSC subscale interaction was not significant (F_10,1125_ = 1.31, *p* = 0.224, ηp^2^ = 0.01). 

Regarding differences among groups on the CBCL broadband scales scores, a significant main effect of the Group was found (F_2,225_ = 16.25, *p* = 0.001, ηp^2^ = 0.13), with lower scores (Tukey’s HSD tests) in children with SLD than in children with ADHD (*p*= 0.001) and children with ADHD+SLD (*p* = 0.001), while children with ADHD and children with ADHD+SLD did not differ (*p* = 0.214). A significant main effect of the CBCL broadband scale was also found (F_1,225_ = 13.93, *p* = 0.001, ηp^2^ = 0.05) with scores (Tukey’s HSD tests) of the internalizing problems subscale significantly higher than scores of the externalizing problems subscale (*p* = 0.001). In addition, the Group × CBCL broadband scale interaction was significant (F_2,225_ = 15.89, *p* = 0.001, ηp^2^ = 0.012). Post hoc analyses (Tukey’s HSD tests) showed that scores of internalizing problems were significantly lower in children with SLD than in children with ADHD+SLD (*p* = 0.019), whereas scores of externalizing problems were significantly lower in children with SLD than in children with ADHD (*p* = 0.001) and with ADHD+SLD (*p* = 0.001). Children with SLD and children with ADHD did not differ on internalizing problems scores (*p* = 0.989), and children with ADHD and with ADHD+SLD did not differ in externalizing problems scores (*p* = 0.998).

The analysis of the correlation between scores of SDSC subscales and of CBCL broadband scales (see Table 3) showed that both internalizing problems and externalizing problems scores were positively correlated (after Bonferroni correction) with SDSC subscales scores with higher scores in the broadband scales corresponding to higher scores in SDSC subscales. Only the externalizing problems scores were not correlated with NH subscale scores.

In order to take into account the possible influences of emotional and behavioral symptoms in comparing the three groups on sleep disturbances, group differences in SDSC subscales were tested controlling for CBCL broadband scale scores. A significant main effect of the Group was confirmed (F_2,223_ = 3.84, *p* = 0.023, ηp^2^ = 0.03), with higher scores in children with ADHD+SLD than in children with ADHD (*p* = 0.003) and with SLD (*p* = 0.001). In addition, the main effect of SDSC Subscale was still significant (F_5,1115_ = 2.68, *p* = 0.021, ηp^2^ = 0.01). The Group × SDSC subscale interaction was not significant (F_10,1115_ = 0.81, *p* = 0.614, ηp^2^ = 0.01), as in the previous analysis without covariates.

## 4. Discussion

The present study aimed to gain insight into sleep disturbances in ADHD and SLD and to better examine sleep disturbances in ADHD and SLD in comorbidity. Our results showed that sleep difficulties were more prevalent in children with ADHD and SLD in comorbidity than in children with ADHD or SLD alone. These sleep difficulties were not associated with differences in age (groups did not differ in age) or differences in behavioral and emotional problems (results were confirmed even after controlling for internalizing and externalizing symptoms).

The critical role of sleep on cognitive functioning has emerged in numerous studies. In children and adolescents who had undergone sleep education programs, improvements in sleep quality were associated with improvements in cognition and academic performance [48,49]. Indeed, high-quality sleep can promote memory consolidation and learning because during sleep, new memories are strengthened and become more resistant to interference [50,51]. On the other hand, in children [52] and adolescents [53,54], minimal but repeated sleep restrictions or sleep deprivation throughout the night have resulted in various cognitive deficits. More specifically, disturbed sleep may lead to slower responses and more variable performance during alertness, vigilance, and attention tasks [55,56]. Additionally, disturbed sleep can increase impulsive errors during executive function tasks [57]. Precisely because sleep plays this crucial role in cognitive functioning, sleep disturbances associated with neurodevelopmental disorders may exacerbate cognitive symptoms and should therefore be carefully studied.

To assess sleep disturbance in children with ADHD and SLD, we opted for the SDSC questionnaire [32]. This instrument can be easily and briefly completed by parents, has good internal consistency and test–retest stability [32], and is among the few sleep questionnaires that meet fundamental operational principles [58]. It provides a total score and six subscale scores (see the materials and methods section for details), which can be used to identify a “personal sleep disturbance profile” and indicate whether there are dysfunctional areas that require further investigation [32]. For these reasons, it has already been widely used to assess sleep difficulties in various clinical populations (e.g., children with cerebral palsy [59]; children with epilepsy [60,61]; obese adolescents [62]; youth with juvenile idiopathic arthritis [63]), including specifically ADHD and SLD [18,22,62,64,65,66]. One of the previous studies using the SDSC in children with ADHD showed that they had difficulty initiating and maintaining sleep, sleep–wake transition, and excessive sleepiness [18]. Recently, the SDSC was used in children with ADHD to evaluate the effect on sleep of mindfulness-oriented meditation training, with positive effects in initiating and maintaining sleep [65]. In addition, the SDSC was used in children with ADHD to assess the impact on sleep of home confinement during the COVID-19 pandemic and was found to reinforce maladaptive sleep patterns during this period [66].

In children with SLD, studies using the SDSC are very sporadic and have yielded mixed results: Some studies have documented problems with initiating and maintaining sleep, sleep breathing, and arousal [64], while others have found no difficulties [22].

According to our findings, comorbidity between ADHD and SLD exacerbates sleep disorders. Specifically, the number of children with ADHD and SLD comorbidities who had clinical levels of disorders in initiating and maintaining sleep, disorders of arousal, sleep–wake transition disorder, and disorders of excessive somnolence was higher than that of children with SLD. In addition, the number of children with ADHD and SLD comorbidity and clinical levels of sleep–wake transition disorder and nocturnal hyperhidrosis exceeded that of children with ADHD. Since comorbidity between ADHD and SLD can be as high as 76% [30], the inconsistencies between previous studies could be due to a lack of attention to this aspect.

Our findings indicate that sleep disturbances should not be overlooked, especially when there is comorbidity between ADHD and SLD. It is important to take this into account because of the role that sleep disturbances have on memory, learning, attention, or executive functioning. In children with ADHD, sleep difficulties have been found to be associated with lower attention skills, specifically the ability to remain alert and vigilant to numerous incoming stimuli [67,68,69], and reduced executive functioning [70]. In addition, in children with ADHD, sleep difficulties have been related to problems with working memory [71] and nighttime consolidation of declarative memories, emotional memories, and socially relevant stimuli [72,73,74]. In children with SLD, the effects of sleep difficulties on cognition are less well known. In a recent study [22], no relationship was found between sleep parameters such as slow-wave activity and novel word recall in children with reading difficulties, unlike in typical readers [23]. However, a relationship between reduced sleep spindles and difficulty in recall has been found in children with reading difficulties [22].

In addition, it is critical to understand how sleep disturbances interfere with the effectiveness of treatments for ADHD and SLD [75]. To date, there are preliminary data showing that in children with ADHD, the positive effect of long-acting stimulant medications on different abilities (i.e., alertness, executive functioning, working memory, and academic productivity) is limited by reduced sleep duration [76].

However, brief behavioral sleep therapy can result in wide-ranging benefits, such as reduced sleep difficulties, alongside improvement in ADHD symptoms, classroom behavior, and working memory tasks [71,77]. Similarly, melatonin treatment may result in a benefit on sleep and ADHD symptoms [78,79,80]. Improving children’s sleep could enhance the improvements due to treatments, with benefits for children and their families, and systematic screening for sleep difficulties with its related intervention should be an integral part of the multimodal treatment plan for children with ADHD [75]. In this regard, improving children’s sleep could enhance treatment-related improvements, benefiting children and their families, and systematic screening for sleep difficulties with its associated intervention should be an integral part of the multimodal treatment provided for children with ADHD [75].

Contrary to previous reports [9,26,27], sleep disturbances in ADHD may not result from emotional and behavioral symptoms alone. According to our findings, in the case of comorbidity between ADHD and SLD, sleep disturbances are not solely dependent on externalizing or internalizing problems and should be specifically treated to improve the quality of life of children and their families [81].

Our study has some limitations. The main one is the lack of a control group. Comparison with a control group could help clarify whether and how much sleep disturbance is exacerbated by the presence of a neurodevelopmental disorder.

Another limitation is the lack of objective measures to assess sleep such as polysomnography and actigraphy. Although evidence supports the validity and reliability of sleep questionnaires, particularly the SDSC [58], future studies with objective measures may be useful in supporting our findings from parent reports [82].

Another aspect that was not assessed in our study was cognitive measures, such as memory and executive functioning. This would allow us to study how any difficulties in memory, learning, and attention are associated with sleep problems in the neurodevelopmental disorders we examined.

Future studies are needed to assess the effects on sleep disturbances of other potentially crucial factors, such as drug treatment and socioeconomic status.

Finally, future longitudinal investigations would also be needed to explore how sleep disturbances may have long-term consequences for neurodevelopmental disorders.

## 5. Conclusions

Our results showed that ADHD and SLD are associated with sleep disturbances and that when the two disorders coexist, sleep disturbances are greater. In conclusion, this result confirms and highlights how complex the relationship between neurodevelopmental disorders and sleep disorders is. Furthermore, this relationship might be bidirectional [11,12,13]. Neurodevelopmental disorders, especially if in comorbidity, might be considered a risk factor for sleep alterations. On the other hand, sleep alterations might lead to emotional, behavioral, and cognitive symptoms that mimic or aggravate neurodevelopmental disorders [2,3,4]. As a consequence, it is possible to hypothesize that similar biochemical disturbances underlie both neurodevelopmental disorders and sleep alterations; for example, a dysfunction in arousal mechanisms might be related to the etiology of both ADHD symptoms and sleep difficulties [11,12,13].

Understanding this relationship is important when assessing and managing patients with ADHD, SLD, and the two comorbid conditions to better understand their difficulties and develop tailored interventions while also paying attention to sleep difficulties.

## Figures and Tables

**Figure 1 ijerph-19-06411-f001:**
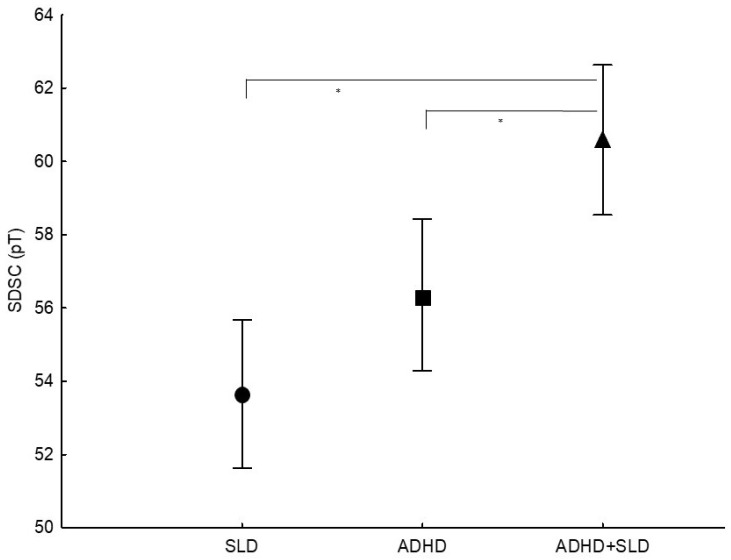
Comparisons of the three groups on the subscale scores of the Sleep Disturbances Scale for Children (SDSC). Children with ADHD and SLD in comorbidity had higher scores on the SDSC subscales than children with ADHD and children with SLD. * *p* ≤ 0.01.

**Table 1 ijerph-19-06411-t001:** Frequencies of children and adolescents with clinical (T-scores ≥ 70) and non-clinical (T-scores < 70) scores and group comparisons on Sleep Disturbances Scale for Children.

	ADHD+SLD		ADHD		SLD		ADHD+SLD vs. ADHD				ADHD+SLD vs. SLD				ADHD vs. SLD			
	M (SD)	c/nc (#)	M (SD)	c/nc (#)	M (SD)	c/nc (#)	OD	95% CI	Z	*p*-Value	OD	95% CI	Z	*p*-Value	OD	95% CI	Z	*p*-Value
DIMS	67 (14.91)	33/43	63 (14.88)	24/50	58 (14.92)	15/63	1.59	0.82–3.19	1.38	0.17	3.22	1.56–6.64	3.17	0.01	2.02	0.96–4.24	1.85	0.06
SBD	56 (12.64)	11/65	53 (12.64)	4/70	51 (12.63)	4/74	2.96	0.89–9.77	1.78	0.07	3.13	0.95–10.31	1.88	0.06	1.06	0.25–4.39	0.08	0.94
DA	60 (15.52)	25/51	57 (15.48)	16/58	56 (15.54)	11/67	1.78	0.85–3.69	1.54	0.12	2.99	1.34–6.63	2.69	0.01	1.68	0.72–3.91	1.21	0.23
SWTD	64 (14.12)	29/47	58 (14.19)	16/58	52 (14.13)	11/67	2.24	1.09–4.61	2.19	0.03	3.76	1.71–8.26	3.29	0.01	1.68	0.72–3.91	1.21	0.23
DES	61 (13.77)	21/55	55 (13.76)	11/63	53 (13.78)	8/70	2.19	0.97–4.94	1.88	0.06	3.34	1.37–8.12	2.66	0.01	1.53	0.58–4.04	0.85	0.39
NH	55 (11.68)	13/63	52 (11.71)	3/71	51 (11.75)	5/73	4.88	1.33–17.93	2.39	0.02	3.01	1.02–8.92	1.9	0.05	0.62	0.14–2.68	0.64	0.52

DIMS = Disorders in initiating and maintaining sleep; SBD = sleep breathing disorders; DA = disorders of arousal; SWTD = sleep–wake transition disorder; DES = disorders of excessive somnolence; NH = nocturnal hyperhidrosis; ADHD = attention deficit hyperactivity disorder; SLD = specific learning disorder; M = mean; SD = standard deviation; c = clinical score; nc = non-clinical score.

**Table 2 ijerph-19-06411-t002:** Means, standard deviations in SDSC subscales, and *p*-values of their comparisons.

SDSC Subscales	M (SD)	DIMS	SBD	DA	SWTD	DES
DIMS	62.76 (15.25)					
SBD	53.29 (12.78)	0.001				
DA	57.73 (15.58)	0.001	0.001			
SWTD	57.91 (14.83)	0.001	0.001	0.999		
DES	56.41 (14.11)	0.001	0.043	0.819	0.724	
NH	52.95 (11.77)	0.001	0.999	0.001	0.001	0.016

SDSC = Sleep Disturbances Scale for Children; DIMS = disorders in initiating and maintaining sleep; SBD = sleep breathing disorders; DA = disorders of arousal; SWTD = sleep–wake transition disorder; DES = disorders of excessive somnolence; NH = nocturnal hyperhidrosis; M = mean; SD = standard deviation.

**Table 3 ijerph-19-06411-t003:** Pearson correlation coefficient (r) and *p*-values for correlation between Sleep Disturbances Scale for Children subscales and CBCL broadband scales scores.

		DIMS	SBD	DA	SWTD	DES	NH
Internalizing Problems	Pearson’s r	0.362	0.246	0.378	0.369	0.447	0.215
	*p*-values	0.001	0.001	0.001	0.001	0.001	0.001
Externalizing Problems	Pearson’s r	0.379	0.248	0.203	0.316	0.403	0.132
	*p*-values	0.001	0.001	0.002	0.001	0.001	0.047

DIMS = disorders in initiating and maintaining sleep; SBD = sleep breathing disorders; DA = disorders of arousal; SWTD = sleep–wake transition disorder; DES = disorders of excessive somnolence; NH = nocturnal hyperhidrosis.

## Data Availability

The data presented in this study are available on request from the corresponding author. The data are not publicly available due to privacy and ethical restrictions.

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
