# Peer review of "Sleep Disturbances in Children with Attentional Deficit Hyperactivity Disorder and Specific Learning Disorders"

_ijerph, 2022, doi:10.3390/ijerph19116411_

Round 1

Reviewer 1 Report

This study aimed at investigating sleep problems in children with attention deficit hyperactivity disorder, specific learning disorder, or both.

I would like to suggest a few modifications.

  1. Please elaborate more on sampling, the period of time patients visited the clinic, and the main complaints that led them to the clinic.
  2. The inclusion criteria 2,3 and 4; are comorbidities that could bias the study results and should be considered exclusion criteria.

Reviewer 2 Report

The important relationship between Deficit Hyperactivity Disorder (ADHD) and sleep disturbances has been increasingly studied.

Significant associations are usually found between sleep disorders and pharmacotherapy and comorbidity.

In this study, this relationship was studied in patients with ADHD and Specific Learning Disorder (SLD), but not compared to a healthy control group. The main outcomes were studied using the Sleep Disorders Scale for Children, and behavioral/emotional problems were also studied, but not associated with confounders such as stimulant drugs for ADHD and socioeconomic status. Consequently, this is an important study but with some flaws.

It would be better if you considered association with confounders such as stimulant drugs for ADHD and socioeconomic status, as already mentioned, considered as a flaw.

Reviewer 3 Report

The authors make passing reference in the discussion to the “bidirectional” nature of sleep disorders and ADHD and Learning disorders but do not cite the literature, which is out there, finding that at least some cases of presumed ADHD may, in fact, be a primary sleep disorder with the ADHD secondary to the poor sleep, with evidence of improvement in ADHD with improved sleep.  I recommend that the bidirectional nature be discussed in the introduction and a bit more prominently with citation of some of that literature.  

Further:

In the last paragraph the authors state: “The relationship between neurodevelopmental disorders and sleep alterations is complex and bidirectional.” Yet the introduction and discussion overwhelmingly focuses on ADHD as the cause of the sleep symptoms rather than the other way around. There is mention of a studies showing that poor sleep can cause cognitive and executive function deficits but there are studies in which ADHD type symptoms are caused by poor sleep disorder and the authors should discuss the idea that children with ADHD type symptoms who meet DSM criteria for ADHD may have a primary sleep disorder accounting for those symptoms rather than just exacerbating those symptoms.
